# Integration of cervical cancer screening into healthcare facilities in low- and middle-income countries: A scoping review

**Rahel Nega Kassa** [1,2] *, **Desalegn Markos Shifti** [3], **Kassahun Alemu** [4], **Akinyinka O. Omigbodun** [5]

**1** Pan African University Life and Earth Sciences Institute (including Health and Agriculture), University of Ibadan, Ibadan, Oyo State, Nigeria, **2** School of Nursing, St. Paul's Hospital Millennium Medical College, Addis Ababa, Ethiopia, **3** Child Health Research Centre, The University of Queensland, South Brisbane, Australia, **4** Department of Epidemiology and Biostatistics, Institute of Public Health, College of Medicine and Health Sciences, University of Gondar, Gondar, Ethiopia, **5** College of Medicine, University of Ibadan, University College Hospital, Ibadan, Oyo State, Nigeria

\* rahelnega208@gmail.com, rahel.nega@sphmmc.edu.et

**Data Availability Statement:** All the relevant data are within the manuscript itself.

**Funding:** This research was funded by the African Union Commission (AUC), Addis Ababa, Ethiopia

## Abstract

Cervical cancer is a prevalent disease among women, especially in low- and middle-income countries (LMICs), where most deaths occur. Integrating cervical cancer screening services into healthcare facilities is essential in combating the disease. Thus, this review aims to map evidence related to integrating cervical cancer screening into existing primary care services and identify associated barriers and facilitators in LMICs. The scoping review employed a five-step framework as proposed by Arksey and O'Malley. Five databases (MEDLINE, Maternity Infant Care, Scopus, Cumulative Index to Nursing and Allied Health Literature (CINAHL), and Web of Science) were systematically searched. Data were extracted, charted, synthesized, and summarised. A total of 28 original articles conducted in LMICs from 2000 to 2023 were included. Thirty-nine percent of the reviewed studies showed that cervical cancer screening (CCS) was integrated into HIV clinics. The rest of the papers revealed that CCS was integrated into existing reproductive and sexual health clinics, maternal and child health, family planning, well-baby clinics, maternal health clinics, gynecology outpatient departments, and sexually transmitted infections clinics. The cost-effectiveness of integrated services, promotion, and international initiatives were identified as facilitators while resource scarcity, lack of skilled staff, high client loads, lack of preventive oncology policy, territorial disputes, and lack of national guidelines were identified as barriers to the services. The evidence suggests that CCS can be integrated into healthcare facilities in LMICs, in various primary care services, including HIV clinics, reproductive and sexual health clinics, well-baby clinics, maternal health clinics, and gynecology OPDs. However, barriers include limited health system capacity, workload, waiting times, and lack of coordination. Addressing these gaps could strengthen the successful integration of CCS into primary care services and improve cervical cancer prevention and treatment outcomes.

through the Pan African University Life and Earth Science Institute (PAULESI), University of Ibadan, Nigeria (No. 1644 to RNK). The funders had no role in study design, data collection and analysis, decision to publish, or preparation of the manuscript.

**Competing interests:** The authors have declared that no competing interest exists.

## Introduction

Cervical cancer ranks as the fourth most prevalent cancer in women worldwide [1]. There were an estimated 604,127 new cervical cancer cases worldwide in 2020 and, the age-standardized incidence rates were highest in eastern Africa, followed by Southern Africa, and Middle Africa respectively [2]. In 2020, low- and middle-income countries (LMICs) accounted for about 90% of the projected 342,000 deaths from cervical cancer [2, 3].

Cervical cancer is preventable and curable, as long as it is detected early and managed effectively [4]. Quick and accurate CCS programs are critical so that every woman with cervical disease gets the treatment she needs, and avoidable deaths are prevented [5]. However, screening coverage of eligible women in most LMICs is on average 19%, compared to 63% in high-income countries [6]. A wide range of barriers, such as lack of knowledge and awareness of cervical cancer, cultural/traditional and religious factors, and health system barriers to screening, were identified across most LMICs [7].

The World Health Organization (WHO) recommends integrating CCS service into primary care packages at healthcare facilities as a solution [8] to overcome the burden related to cervical cancer in LMICs [9]. Therefore, the availability of integrated screening services in healthcare facilities can be used as an opportunity for those women who visit the facilities for different reasons to be screened and to achieve the 2030 cancer elimination goal (90-70-90 strategy) of the WHO. These are, 90% of women should get vaccinated, 70% of age-eligible women should get screened and 90% of women who have abnormalities detected should get treatment by 2030 [10]. Integrated service delivery leads to better quality of care, greater service provision, higher utilization of sexual and reproductive health services, more efficient use of resources, and better client satisfaction [11–13].

Although several studies [14–19] were conducted on CCS uptake among women in LMICs, there is still room for more comprehensive evidence on integration of CCS into healthcare facilities of LMICs. Therefore, the aim of this scoping review is, to summarize the existing evidence on the integration of CCS into healthcare facilities, and to identify barriers and facilitators associated with the integration of CCS programs into primary care in LMICs.

## Methods

### Review design

The protocol for this review has been published in BMJ Global Health Journal. The review was guided by the scoping review framework proposed by Arksey and O'Malley [20]. The framework consists of five steps such as 1) formulating the research questions, 2) identifying relevant studies, 3) selecting eligible studies, 4) charting the data, and 5) collating, summarising, and reporting the results as described below.

### 1. Formulating the research questions

Our research questions were developed and refined through an iterative process and consultations held by the multidisciplinary research team. The following questions were addressed by this scoping review: i) what is the existing evidence on the integration of cervical cancer screening programs into healthcare facilities in LMICs? and ii) what are the barriers and facilitators associated with the cervical cancer screening integration in healthcare facilities in LMICs?

### 2. Identifying relevant studies

The search strategy included terms that covered healthcare facilities, integration of cervical cancer screening, and LMICs (S1 Table). The search strategy was piloted to ensure the

appropriateness of proposed keywords and databases. Peer-reviewed literature published were retrieved from the year 2000, which was a starting point for the implementation of the Millennium Development Goals (MDGs) that included achieving universal access to reproductive health by 2015 [21], till the year 2023.

A comprehensive literature search was conducted on the following five electronic databases: MEDLINE, Maternity Infant Care, Scopus, Cumulative Index to Nursing and Allied Health Literature (CINAHL), and Web of Science. Manual searches of the articles' reference lists were also conducted to identify additional potentially eligible studies not found in the databases.

## 3. Selecting eligible studies

**Inclusion criteria.** The eligibility criteria was set based on population, concepts, and contexts (PCC) framework described by JBI which was proposed by Peters *et al.* [22], shown in Table 1. Regarding the study type, we plan to include all primary studies published in a peer-reviewed journal that is accessible online and through interlibrary requests. Such studies could include randomized trials, observational studies, cross-sectional studies, case studies, and laboratory studies. Articles with non-English language and non-journal articles will be excluded.

**Screening.** Eligibility screening was started by screening the title and abstracts of the included studies. All eligible articles were uploaded into Endnote 20 reference management software [24] and duplicates were identified and removed. Titles and abstracts screening and full-text reviews were done independently by two researchers [RNK and DMS] using Covidence software and checked the agreement of the included studies. A third reviewer [25] was employed to resolve the disagreement that were not resolved by discussion and consensus.

From April to September 2023, we conducted searches and independently evaluated the titles and abstracts of relevant publications using specific criteria for inclusion and exclusion. Records(n = 10,476) from the indexed articles were compiled, and duplicates were removed (n = 3,666). We screened records (n = 6,810) at the title and abstract screening level that resulted in (n = 120) articles that were screened for the full text review. Finally, 28 articles were included in the review process (Fig 1).

**Table 1. Inclusion criteria using the PCC framework.**

| Criteria | Description |
|---|---|
| P-Population | • Specialised Hospitals<br>• General Hospitals<br>• Primary Hospitals<br>• Central Hospitals<br>• Health Centres<br>• Clinics<br>• Primary care clinics<br>• Primary care services (MCH, HIV service, TB service, NCDs clinic, Outpatient clinic Reproductive health services, FP, routine genecology service) |
| C-Concept | • Cervical cancer screening integration<br>• Barriers and facilitators of cervical cancer screening integration |
| C-Context | Studies:<br>• Conducted in low and middle-income countries [23].<br>• Published from 2000 to 2023<br>• Written in English<br>• All primary studies<br>✓ Quantitative (cross-sectional/ observational),<br>✓ Qualitative (phenomenology, case study) and,<br>✓ Mixed-method published articles). |

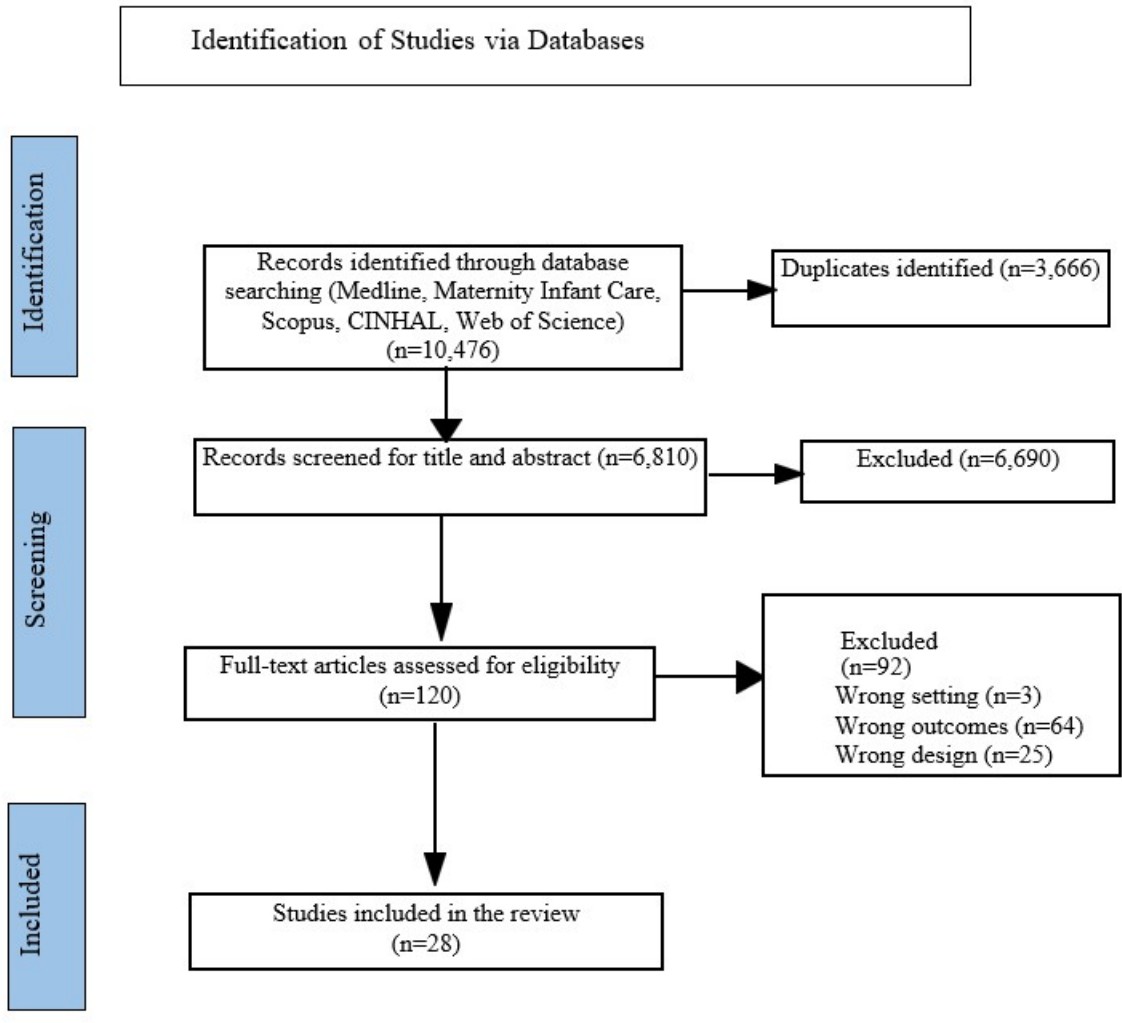

**Fig 1. PRISMA-ScR flow chart illustrating the process of literature selection.**

## 4. Charting the data

A spreadsheet of key factors was adapted and developed to extract relevant data from each included study [26, 27]. Two authors [RNK and DMS] tested and refined the data extraction tool using 10 eligible studies before its use. Reviewers extracted data from all eligible articles using the final form. The extracted data included key variables such as i) author(s) and year of publication, ii) country iii) study design iv) study period v) study setting, vi) study participants, vii) data collection methods, viii) key findings and ix) limitations of the study (Table 2).

## 5. Collating, summarising, and reporting the results

Findings of the scoping review were presented using the Preferred Reporting Items for Systematic Reviews and Meta-Analyses Extension for Scoping Reviews (PRISMA–ScR) guidelines [28]. A PRISMA flow diagram was used to demonstrate the literature study selection process and search results. A descriptive numerical summary was used to present the characteristics of the included studies while a qualitative thematic synthesis was conducted to narrate the findings. The results were classified under the main conceptual categories, such as "integration of

**Table 2. Summary of the articles.**

| Authors | Country | Study type/design | Study period | Study participants | Study sample | Study setting | Data collection methods | Limitations |
|---------|---------|-------------------|--------------|--------------------|--------------|----------------|--------------------------|-------------|
| Claey P et al. 2003 [45] | Kenya | Mixed design | Nov1999 to Feb 2000 | Women attending family planning clinics, health care professionals | 981 | Family Planning Association of Kenya (FPAK). | Interviews and, a review of patient files | ————— |
| Were E et al. 2010 [46] | Kenya | Cross-sectional | May 2005 to Jan. 2006 | Women | 435 | MTRH, MCH-FP clinic, and well-baby clinic in Teaching and Referral Hospital | Face-to-face interviews | ————— |
| Megan J et al.2015 [52] | Kenya | Cross-sectional | Oct 2007 to Oct 2010 | Health care providers, women aged less than 25 | 418 | HIV care and treatment clinics at two hospitals | Document review | ————— |
| Smit et al. 2012 [47] | South Africa | Qualitative | ————— | Policymakers, program managers, and academicians working in health policy | 21 | HIV clinic | In-depth interview | Risk of subjectivity |
| Odafe, S. et al. 2013 [53] | Nigeria | Cross-sectional | Apr2010 to Apr2011 | Women living with HIV | 805 | RH services clinic in district hospital | EMR | ————— |
| Padmaja R et al. 2013 [54] | India | Cross-sectional | Two years | Women aged 19 and above | 350 | (any clinic) in tertiary Hospital | Specimen collection | Small sample size |
| Chawla PC et al.2014 [51] | India | Cross-sectional | Aug 2012 to Apr 2013 | Public hospitals, Private Hospitals, and Primary health center (PHC) | 312 | Public hospitals, Private Hospitals and PHCs | Standardized questionnaire | ————— |
| Elizabeth Roger et al. 2014 [49] | Haiti | Qualitative/case report | ————— | A 50-year-old female | 1 | Maternal Health Clinic | Interview | ————— |
| Kumakech et al. 2014 [38] | Uganda | Qualitative study | Feb 2012 to Feb 2013 | HCPs, Policymakers | 16 | Health facilities with HIV & CCS services clinics | In-depth interview | Small sample size, risk of response bias, risk of subjectivity |
| Anderson J et al.2015 [44] | Côte d'Ivoire, Guyana, and Tanzania | Cross-sectional | Jan 2009 to Mar 2012 | Women aged 30 to 50 years but not limited to HIV-positive women | 3,4921 | HIV clinics, RCH clinics in National hospitals, regional hospitals, District hospitals, and Health centres | Data extraction from individual client records | ————— |
| EL Sibanda et al. 2015 [37] | Zimbabwe | Qualitative | Nov.2013 to Mar.2015 | Women aged 18 to 52 | 69 | Clinics providing integrated services | In-depth interviewa | ————— |
| Kumakech et al. 2015 [39] | Uganda | Qualitative study | Feb. 2013 to Jan. 2014 | Women and village health teams, husbands | 24 | 3district hospitals | FGDs and IDIs | ————— |
| Shiferaw et al. 2016 [55] | Ethiopia | Cross-sectional | Aug. 2010 to Mar. 2014 | Women with HIV | 16,632 | HIV clinics in secondary and tertiary health facilities | Document review | Lack of generalizability |

(Continued)

**Table 2.** (Continued)

| Authors | Country | Study type/design | Study period | Study participants | Study sample | Study setting | Data collection methods | Limitations |
|---|---|---|---|---|---|---|---|---|
| Bekolo CE et al.2016 [34] | Cameroon | Cross-sectional | Feb and May 2014 | Women aged 35 years and above attending HIV clinic | 302 | HIV clinic in a Regional Hospital | Interview using a questionnaire | ——— |
| Elisabeth L. Vodicka et al., 2017 [42] | Kenya | Cross-sectional | July 1 to Oct 31, 2014 | Women aged 18 and above, clinic staff | 319 | Child health clinic | interviews | ——— |
| Jenell S. et al. 2016 [56] | Sub-Saharan African countries | Cross-sectional | Feb to Dec 2013 | HIV clinics | 78 | HIV clinics | Online interview, and patient record review. | Sampling bias, low response rate, lack of representativeness |
| Katie A. Ports et al. 2015 [41] | South Africa | Mixed design | ——— | Women aged 18 and above | 79 | Homeopathic clinic | Face-to-face interview, IDIs | ——— |
| Rupani, M. P., et al. 2017 [50] | India | Cross-sectional | June 2011 to Oct. 2011 | Reproductive age women | 356 | Gynecology OPD in tertiary care teaching hospital | Interview | Small sample size/lack of generalizability, response bias |
| Sam Phiri et al. 2016 [40] | Malawi | Cross-sectional | Feb 2012 of May 2014 | Reproductive age women | 1495 | FP and RH clinic | Specimen collection | ——— |
| Wendimagegn NF.2019 [57] | Ethiopia | Mixed design | ——— | Women | 296 | Any clinic in hospitals and health centres | Interview | Lack of representativeness |
| Cecilia Milford et al, 2018 [58] | South Africa | Mixed design | 2009 to 2011 | Healthcare providers, senior management, professional nurses and doctors, counsellors | 86 | 1 District hospital, 5 PHC, and one community health centre | Focus group | ——— |
| Colin Pfaf et al. 2018 [35] | Malawi | Cross-sectional | May2016 to Mar.2017 | Women aged 20 and above years | 957 | HIV clinic in Central Hospital/District | Extracted from Electronic Medical Record [59], | ——— |
| Boddu A et al. 2021 [60] | India | Cross-sectional | Mar2014 to Feb2015 | Women aged 30 to 59 | 404 | HIV clinics | Face-to-face Interview | ——— |
| Mpata PC et al. 2021 [61] | Zimbabwe | Qualitative/ phenomenological | Nov. 2017 to Feb. 2018 | Women living with HIV | 36 | Opportunistic infection clinics in the hospitals | IDI, FGDs | ——— |
| Prisca C et al. 2021 [62] | Kenya | Mixed design | June 2020 to Aug2020 | Healthcare providers | 79 | 20 PHC | Face-to-face interview | Small sample size, lack of generalizability, sampling error |
| Farida Selmouni et al. 2022 [48] | Benin, Cote d'Ivoire, and Senegal | Quantitative/ observational | May 2018 to Jan. 2021 | Women aged 25–4 | 16,530 | Primary health centres | Performance checklist | Sampling bias |
| Akpan, E et al. 2023 [63] | Kenya | Qualitative | ——— | Nurses and managers | ——— | Primary health facilities | In-depth interview | Lack of generalizability |

*(Continued)*

**Table 2.** (Continued)

| Authors | Country | Study type/design | Study period | Study participants | Study sample | Study setting | Data collection methods | Limitations |
|---|---|---|---|---|---|---|---|---|
| Ninsiima et al. 2023 [25] | Uganda | Mixed design | ————————— | HIV-infected women | 327 | HIV clinic in Regional Referral Hospital in Mbarara District | Face to face interview, FDGs | Lack of generalizability |

CCS: Cervical Cancer Screening; CHC: Child Health Clinic; EMR: Electronic Medical Record; FPAK: Family Planning Association of Kenya; FGD: Focus Group Discussion; IDI: In depth Interview; HCPs: Health care providers; HIV: Human Immunodeficiency Virus; PHC: Primary Health centre; MCH: Maternal & Child Health Clinic; MTRH: Maternal &Reproductive Health; OPD: Outpatient department; RCH: Reproductive & Child Health; RH: Reproductive Health

cervical cancer screening" and "facilitators and barriers to the integration program". Under each category, we further provided data on the article's characteristics, including, but not limited to, the total number of studies, types of study design, sources of data, year of publication and key findings. Tables and figures were used to present the results in line with the aims of the scope of the review.

**Quality assessment.** Joanna Briggs Institute (JBI) quality appraisal tools [29–33] were used to assess the methodological quality of the included articles by evaluating the extent to which they addressed the possibility of bias in areas of study design, conduct, and analysis (S2 Table). The magnitude of studies with a total score indicating poor quality was discussed among the researchers. Two researchers [RNK and DMS] independently assessed each included paper and any uncertainty regarding the quality of publications was resolved through discussion. Although a formal assessment of the methodological quality of the included studies was performed; articles with poor quality were not excluded.

Accordingly, we evaluated a total of 21 cross-sectional research articles and 7 qualitative research articles. Out of the 21 cross-sectional studies, only one achieved a perfect score of 8 out of 8. Five of the studies received a score of 6 out of 8, while 5 studies scored 5 out of 8. Two articles failed to identify and address confounding factors, while another three articles were unclear in their strategies for dealing with them. Additionally, three studies did not provide clear criteria for sample inclusion, and three studies did not properly state the validity and reliability of their exposure measurements.

Among the 7 qualitative research articles, 3 achieved a score of 8 out of 10, 2 received a score of 7 out of 10, and the remaining 2 scored 6 out of 10. Besides, all included articles drew their conclusions from data analysis or interpretation. Over half of the articles did not explicitly state the cultural and theoretical background of the researcher. Five articles did not address the influence of the researcher on the research, while four articles did not adequately represent the study participants and their perspectives.

## Results

### Characteristics of the reviewed articles

A total of 28 articles were summarized in this review (Table 2), which were obtained from indexed databases. These studies covered a range of low- and middle-income countries (LMICs), including Kenya, India, Uganda, South Africa, Ethiopia, Malawi, Zimbabwe, Benin, Cote d' Ivoire, Senegal, Cameroon, Guyana, Tanzania, Nigeria, and other sub-Saharan African countries. Of the articles reviewed, over half (53.6%) were quantitative studies. In addition to these studies, there were also six articles with a mixed study design and seven articles with a qualitative research design. A significant portion (26.6%) utilized patient records and analysed data from health system repositories. As a result, the sample sizes of these studies varied widely, with a range from 1 participant to as many as 34,921, and a median sample size of 319.

The majority (57.1%) of the included studies were conducted in hospital settings, such as referral hospitals, regional hospitals, district hospitals, and one private hospital. Only six studies focused on primary healthcare settings. The studies covered a range of clinics, with the plurality (39%) focusing on HIV clinics. Others included child health clinics, maternal health clinics, family planning clinics, reproductive health clinics, integrated service clinics, opportunistic infection clinics, and homeopathic clinics. In most studies, CCS integration mainly occurred in HIV clinics [25, 34–40]. A study conducted in South Africa showed that all key informants were in support of providing pap smears at HIV clinics because they saw a need for healthcare to be more comprehensive [41, 42].

However, some studies found that CCS was integrated into existing reproductive and sexual health [43] clinics, MCH-FP, well-baby clinics, maternal health clinics, Gynecology OPD, STD clinics and this was feasible in resource-poor settings [44–50]. Therefore, with adequate infrastructural support, training, mentoring, and program supervision, cervical cancer screening and treatment services can be effectively provided opportunistically [48].

More than two-thirds (67.8%) of the studies included women aged 18 to 59 as study participants, whereas the other ones considered policy makers, senior management, healthcare providers, academicians working in health policy, counsellors, village health teams, and husbands. Two studies even utilized the health facilities themselves as study participants, specifically the HIV clinics and a combination of public hospitals, private hospitals, and primary care centres. The frequently used data collection methods were face-to-face interviews, focus group discussions and in-depth interviews.

## Cervical cancer screening integration into the healthcare facilities in LMICs

Through this review, we identified ways of CCS integration into healthcare facilities and explored different facilitators and barriers to the integration of CCS, which are detailed in (Tables 3 and 4). The integration of CCS into the health care facilities was thematized into two groups: *ways* of integrating the CCS into the health care facilities, and *benefits/acceptance* of integrated CCS service.

## Ways/sites of integrating ccs into healthcare facilities

In most studies, CCS integration mainly occurred in HIV clinics [25, 34–40]. A concurrent mixed design study conducted in South Africa showed that all key informants were in support

Table 3. Key findings on the integration of CCS service into health facilities in LMICs.

| Theme | Conclusion |
|---|---|
| *Ways/sites of Integrating the CCS into healthcare facilities* | |
| CCS integration mainly occurred in HIV clinics[25, 34–42]. CCS was also integrated into existing reproductive and sexual health [43] clinics, MCH-FP, well-baby clinics, maternal health clinics, Gynecology OPD, STD clinics [44–50, 53]. | Integrated cervical cancer screening and treatment services can be effectively provided opportunistically and feasible in resource-poor settings [36, 48, 53, 54]. |
| *Acceptance of integration of CCS service into healthcare facilities* | |
| Most of the women who participated in the studies accepted cervical cancer screening integrated within HIV clinics, the MCH-FP clinics, and sexual and reproductive health [25, 40, 46, 53, 64]. | Integration of cervical cancer screening and genital tract infection identification and treatment into the existing MCH-FP appears feasible [46]. |
| *Benefits of integration of CCS service into health care facilities* | |
| Comprehensive and convenient to access all services under one roof, facilitates early detection and treatment, minimizes loss to follow-up, cost-effective, and perceived benefits, and reduces patient visits [25, 37, 38, 40–42, 45, 61]. | CCS could be integrated into other existing healthcare services, such as routine immunizations and family planning programs, where women already engage with the healthcare system [38, 49]. |
| *Disadvantage of integrated CCS services* | |
| Increases the workload for healthcare providers, and fragmentation of services if the service is given in a separate room within the health facility [62]. Feeling empathy/emotional distress for testing positive for both HIV and cervical cancer lesions, decrease in the number of women screened per day, and feeling disrespectful or offensive to their needs due to the prolonged time [25, 37, 38, 45, 58]. | |

**Table 4. Key findings on facilitators and barriers of integrated cervical cancer screening service in LMICs.**

| Theme/ Level | Facilitators | Barriers | Possible solutions |
|---|---|---|---|
| Individual/ Women-Level | Comprehensiveness of the services, accessibility of multiple services [37, 45]. Willingness to undergo screening [48]. Perceived risk, support from peers, perceived risk, availability of free services, and increased spread of information [37]. Cost-saving approach [42]. | Difficulties in persuading women, and reluctance to visit another clinic if the integration is in a separate clinic. Economic constraints and lack of social support, long waiting times [25, 45, 53, 60, 63]. Lack of awareness [36, 45, 57]. Perceived barriers, and perceived severity [37, 45, 52]. | Organizing information and education sessions on cervical cancer, and taking initiatives to reduce financial barriers [42, 60]. |
| Healthcare Provider-Level | Dedication and enthusiasm of the healthcare workers in advising and encouraging women. Adherence to the screening guidelines [48]. | A lack of staff, insufficient on-the-job training, poor interpersonal care and counselling skills, and resistance to change [47, 62, 64]. Increased workloads [48, 55], and the prioritization of curative healthcare by health professionals [57]. | Cost containment of skilled staff [36]. |
| Institutional/ Organizational-Level | Promotion of cervical cancer screening through various methods [45]. Supervisory visit [48]. | A lack of resources, inappropriate use of funds, and competing priorities in ensuring comprehensive and sustainable HIV care [34, 51]. Trained staff turnover and a lack of regular supply of necessary equipment for cervical cancer screening, shortage of medication, protocols, guidelines, and high costs of preventive healthcare services [48, 55, 57]. Shortages of skilled health staff, high client loads, and territorialism among providers and managers. Weak supervision and management, the absence of integration indicators in monitoring and evaluation tools, and the lack of integrated care in performance appraisal, accreditation processes, and job descriptions have further been cited as obstacles, lack of infrastructure [36, 47, 57, 62]. Availability of more than one screening method [63] | Strong advocacy and leadership [48]. Task shifting [63, 66]. |

*(Continued)*

**Table 4.** (Continued)

| Theme/ Level | Facilitators | Barriers | Possible solutions |
|---|---|---|---|
| Policy and Systems-Level | Recognizing the importance of integration, international support for SRH-HIV integration, advocating for decentralization of services, implementing task-shifting protocols, and promoting the concept of "provider-initiated SRH." [47]. | Absence of functional policy on preventive oncology, a lack of action from authorities, and insufficient funding [51]. Verticalized program structures, low funding and attention to SRH nationally, inadequate coordination between different levels of policy, territorial disputes over programs, absence of national guidelines on integration, and poor coordination between different healthcare sites [47]. | Ensuring the implementation of provider-initiated SRH service [47]. Creating a coordinated network [36]. |

of providing pap smears at HIV clinics because they saw a need for healthcare to be more comprehensive [41, 42].

However, some studies found that CCS was integrated into existing reproductive and sexual health [43] clinics, MCH-FP, well-baby clinics, maternal health clinics, Gynecology OPD, STD clinics, and this was feasible in resource-poor settings [44–50]. Therefore, with adequate infrastructural support, training, mentoring, and program supervision, cervical cancer screening and treatment services can be effectively provided opportunistically [48].

## Acceptance of integration of CCS service into healthcare facilities

The study in Kenya found that (87%) of women receiving HIV care and treatment received CCS services. Among the women screened, almost all (96%) accepted screening during the current visit, while the remaining women were screened during a first or second follow-up visit [64].

Similarly, a mixed approach study conducted in Uganda revealed that the majority of HIV-infected women (64.5%) accepted the integration of CCS into routine HIV care [25]. Studies also revealed that patients found integrated sexual and reproductive health (SRH)/Antiretroviral therapy(ART) services acceptable [40]. Another study from Kenya also showed that most of the women included in the study accepted CCS services integrated with the MCH-FP clinics [46].

## Benefits of integration of CCS service into healthcare facilities

Integrating CCS services into existing clinics would provide multiple benefits, including access to more health services in a single visit and increased access to CCS for HIV-positive women. It would, indeed, be convenient to access all services under one roof. Integration was also seen as a way to prevent HIV-positive women from dying from cervical cancer and to reduce the frequency of visits to health facilities. It was also believed that integration would facilitate early detection and treatment of gynaecological diseases. Integration would enable healthcare providers to detect and treat cervical cancer lesions at an early stage, reducing mortality from the disease. Integration was also seen as a way to minimize loss to follow-up in the CCS program and to increase the availability of screening sites for women [25, 41, 61].

A study conducted in Kenya found that integrating cervical cancer screening into HIV clinics would be cost-saving from a societal perspective [41]. This is because integrating the screening program reduces overhead costs by utilizing existing resources [25]. Furthermore, the study showed that offering screening to women at the time of their HIV treatment greatly reduced costs associated with patient transportation and time [38, 42].

The perceived benefits include the convenience of receiving both HIV drugs and CCS on the same scheduled date, reducing disturbance and movement by seeking these services from different clinics. There is motivation to undergo CCS due to increased awareness and understanding of the importance of early detection and treatment of precancerous lesions. Compliance and adherence to annual screening are improved, and there are increased opportunities to undergo screening. Archiving screening results and HIV-related information in one file improves record keeping. Confidentiality of HIV patient information is maintained, reducing stigma from HIV-uninfected women and non-HIV health workers. There is also a preference to interact with HIV clinic health workers as they are familiar and trusted [25].

## Disadvantage of integrated CCS services

Participants in the study expressed concerns regarding the limited health system capacity and the potential consequences of integrating HIV testing and cervical cancer screening. One worry was that women who fear HIV stigma might be deterred from getting screened for cervical cancer if the two are integrated. Additionally, participants raised concerns about the increased workload for healthcare providers and how it might impact efficiency and productivity. Integration could lead to a decrease in the number of women screened per day. Fragmentation of services also could be one of the disadvantages if the service is given in a separate room within the health facility [62].

Another concern that emerged from the study was the feeling of compassion towards the women. Some participants expressed empathy for those who test positive for both HIV and cervical cancer lesions, recognizing the heightened emotional distress they may experience. There was also a concern for the prolonged waiting time that women might face due to the integrated approach, as it could be seen as disrespectful or offensive to their needs [25, 38, 58]. Furthermore, participants felt compassion for women who might struggle to comprehend the integrated format and the concepts of both HIV and cervical cancer screening.

## Barriers and facilitators to integrating cervical cancer screening into health institutions in LMICs

The barriers and facilitators were classified into four categories: individual/women, provider, organizational/system and policy levels.

## Individual-level facilitators and barriers

**Facilitators.** Women were motivated to seek CCS services in family planning clinics due to the service's high quality and comprehensive approach, conveniently providing access to multiple services [37, 45]. The willingness of women to undergo screening and receive treatment also played a significant role in facilitating the provision of these services within primary health centres [48]. Additionally, factors such as personal connections with those affected by cervical cancer, support from peers, experiencing suspicious symptoms, availability of free services, and the spread of information on CCS further contributed to the uptake of these services [37]. It was also highlighted that integrating CCS into HIV clinics is a cost-saving approach, compared to conducting non-integrated screening, when considering societal perspectives [42].

Besides, religion, high perceived risk of developing cervical cancer, and ever-screened for cervical cancer facilitated the acceptability of integration of cervical cancer screening into routine HIV care [25].

**Barriers.** Difficulties in persuading women to undergo CCS at the Gynaecology OPD after their ART Clinic appointment are evident. One major obstacle is their reluctance to visit another clinic after already devoting time to the ART Clinic and going through CD4 tests and documentation. To compound the issue, some women were uncertain about where to go and needed assistance from an attendant. Economic constraints and lack of social support to care for their children while they were at the clinic further hindered their willingness. Additionally, long waiting times deterred some women from seeking screening services [45, 60].

Another study also revealed numerous barriers faced by women, such as lack of awareness, poor knowledge and neglect, all of which hampered their willingness to undergo screening [45, 57]. Furthermore, discomfort in sharing their privacy with health workers who were aware of their HIV status, preferring to confide in health workers who had no prior knowledge of their condition, and apprehension about continued interactions with the same health workers for other HIV-related services presented challenges for integrating CCS into routine HIV care. Increased waiting times at the HIV clinic for CCS, as well as receiving both HIV and CCS services, were also highlighted as hindrances [25].

Other individual-level barriers identified in several studies were perceived patient barriers. Fear of the results and a cancer diagnosis further complicated the situation. Some women perceived CCS as a complex procedure, often referring to it as an "operation," fearing that manipulation of the fragile cervix could potentially cause cancer. Many had a misconception that only older or HIV-positive women were susceptible to cervical cancer and believed that the absence of signs or symptoms indicated a low risk of disease [37, 45, 64].

## Healthcare provider-level facilitators and barriers

**Facilitators.** The key element in increasing women's involvement in CCS was the dedication and enthusiasm of the healthcare workers in advising and encouraging eligible women to undergo screening. The Visual Inspection with Acetic Acid [65] providers adhered closely to the screening guidelines, evident through almost no refusal of treatment by the providers [48].

**Barriers.** While healthcare providers expressed their eagerness to incorporate cervical cancer screening services into the current system, they acknowledged several obstacles. These barriers included fragmented services, a lack of staff, insufficient on-the-job training, and limited patient education [62]. Providers also voiced concerns about increased workloads [48, 55] and the prioritization of curative healthcare by health professionals, which further hindered the integration of these services [57].

## Institutional/Organizational-Level Service Facilitators and Barriers

**Facilitators.** The promotion of cervical cancer screening through various methods such as posters in waiting rooms, direct communication between healthcare providers and patients, and even community-based family planning methods distributors in rural and slum areas was one of the facilitators to increase acceptance of CCS by the women [45]. Additionally, organizing supervisory visits when there is a decrease in the number of women being screened in a specific clinic has consistently led to a higher uptake of CCS [48].

**Barriers.** Several studies have identified various barriers to integrating CCS into HIV care. These barriers include a lack of resources, inappropriate use of funds, and competing priorities in ensuring comprehensive and sustainable HIV care [34, 51]. Additionally, challenges

such as trained staff turnover and a lack of regular supply of necessary equipment for CCS have been reported in some centres. Other identified barriers include a shortage of medication, protocols, guidelines, and high costs of preventive healthcare services [48, 55, 57].

Moreover, shortages of skilled health staff, high client loads, poor interpersonal care and counselling skills, judgmental attitudes, resistance to change, and territorialism among providers and managers have posed challenges. The orientation of HIV services to disease management and a lack of technical skills in sexual and reproductive health have also been identified as barriers. Ineffective training methodologies, weak supervision and management, the absence of integration indicators in monitoring and evaluation tools, and the lack of integrated care in performance appraisal, accreditation processes and job descriptions have further been cited as obstacles. Finally, infrastructure issues, such as limited access to water and electricity, inadequate space, and challenges in the referral system, have all been reported [47, 57, 62, 64].

### Policy and systems-level facilitators and barriers

**Facilitators.**   Some factors have been identified as policy-level facilitators for the integration of sexual and reproductive health (SRH) with HIV services. These include recognizing the importance of integration, receiving international support for SRH-HIV integration, advocating for decentralization of HIV services, implementing task-shifting protocols, and promoting the concept of "provider-initiated SRH." [47].

**Barriers.**   The main reasons for the insufficient facilities and supplies for CCS in health centres were identified as the absence of a policy on preventive oncology, lack of action from authorities, and insufficient funding. Additionally, the lack of a functional policy on preventive oncology and the lack of action from authorities were noted as contributing factors [51]. Other issues included verticalized program structures in the Department of Health, low funding and attention to sexual and reproductive health nationally, inadequate coordination between different levels of policy, territorial disputes over programs, absence of national guidelines on integration, a dysfunctional referral system, and poor coordination between different healthcare sites [47].

## Discussion

The findings of the scoping review on CCS integration into healthcare facilities in LMICs provide valuable insights into the ways, benefits, and acceptability of integrating CCS services into existing healthcare settings. The review identified the sites of integration, the acceptance of the integrated services by women, and the advantages and disadvantages associated with the integration.

One key finding is that most of the CCS integration occurred in HIV clinics, compared to the other sites available for opportunistic intervention. This is consistent with other findings which revealed the integration of cervical cancer screening in HIV clinic [65, 67–70]. This finding highlights the potential benefits of integrating CCS services into HIV clinics, as it allows for streamlined care and ensures that HIV-positive women have access to CCS [65].

On the other hand, the review also revealed that CCS was found to be integrated into other healthcare settings such as reproductive and sexual health clinics, maternal health clinics, well-baby clinics, and gynecology OPDs. This finding is supported by similar studies which showed that CCS service was delivered opportunistically in the existing reproductive health services [71–73]. This implies that with adequate infrastructural support, training, mentoring, and program supervision, CCS can be effectively provided opportunistically in these healthcare settings. This finding also suggests that integration of CCS services is feasible across various healthcare facilities in resource-poor settings. Consistent with reports which revealed that it is

feasible for low-income countries to integrate cervical cancer prevention, screening and treatment into routine women's health services [66, 70, 74, 75], such integration requires political will, cross-sectoral collaboration and planning, innovative partnerships and robust monitoring and evaluation [76].

The overall acceptance of integrated CCS services among women was found to be high in this review. This finding aligns with another similar study that reported a higher acceptance rate of opportunistic CCS among women [71]. It suggests that offering CCS services alongside other healthcare services can enhance acceptance and uptake among women, which is consistent with a scoping review that emphasized the importance of improving availability and accessibility of services to increase CCS uptake among women [77].

Integrated CCS services have numerous benefits, such as improving access to health services, preventing HIV-positive women from dying from cervical cancer, reducing the frequency of visits, facilitating early detection and treatment of gynaecological diseases, minimizing loss to follow-up, improving patient records and confidentiality, and being cost-saving. This is congruent with previously conducted study and recommendations of WHO's cervical cancer elimination guideline for LMICs [9, 10, 65, 70].

During the scoping review, several barriers and facilitators to cervical cancer screening at various levels were also identified. At the individual/women level, several facilitators were identified. Women were motivated to seek cervical cancer screening services in family planning clinics due to the high quality and comprehensive approach offered. Factors such as availability of free services, and spreading information on cervical cancer screening further contributed to the uptake of these services. This is in agreement with similar studies that showed the availability of cervical cancer screening with family planning services and other primary care services facilitated CCS uptake [65, 78, 79]. It was also found that integrating cervical cancer screening into HIV clinics is a cost-saving approach which is in line with a previously conducted studies that revealed integrated CCS services is cost effective in terms of both direct and indirect service costs [65, 73, 76]. These facilitators highlight how providing high-quality services, comprehensive care, and involving individuals and communities in spreading awareness can increase the likelihood of women seeking screening services. This is also supported by another study which explored that involving different stakeholders to deal with integrated screening services would increase CCS uptake [65].

Nevertheless, there are several barriers at the individual/women level that hinder the uptake of cervical cancer screening. Lack of awareness, knowledge, long waiting times, discomfort in sharing privacy, fear of results and cancer diagnosis, perception of screening as a complex procedure, and misconceptions about risk factors further compounded the situation which is in agreement with previous studies [7, 78–80]. Such barriers emphasize the importance of reducing structural and logistical challenges, improving education and awareness, addressing misconceptions, and implementing strategies to minimize waiting times.

At the healthcare provider level, healthcare providers' adherence to screening guidelines, and their advice played a significant role in motivating and encouraging eligible women to undergo screening which is consistent with a study that identified encouraging women to have CCS increases the uptake [81]. However, there were also barriers reported by providers, such as fragmented services, a lack of staff, insufficient training, increased workloads, and prioritizing curative healthcare which is supported by other studies [65, 82, 83]. These barriers shed light on the need to ensure adequate staffing, resources, training, and institutional support for healthcare providers to integrate cervical cancer screening effectively.

At the institutional/organizational level, promoting screening through various methods and organizing supervisory visits were identified as facilitators, and found to increase acceptance of screening services which aligns with previous study [83]. However, several barriers to

integration were also highlighted at this level, including resource constraints, competing priorities, turnover of staff, lack of necessary equipment, medication shortages, guideline adherence, and high costs of preventive care services. These barriers underline the need for increased investment, proper allocation of resources, strong management systems, and improved coordination between different levels of care provision [7, 78].

Finally, at the policy and systems level, recognizing the importance of integration, receiving international support, decentralized HIV services, task-shifting protocols, and provider-initiated sexual and reproductive health initiatives were identified as policy-level facilitators. This is in line with suggestions by the WHO's cervical cancer elimination strategies for LMICs and another study [10, 70]. However, lack of policies, insufficient funding, absence of guidelines on integration, vertical program structures, territorial disputes, dysfunctional referral systems, and poor coordination between healthcare sites and policies were highlighted as barriers. This is in line with similar studies [7, 69, 78] These findings reveal the need for policy changes, improved coordination, and robust support for integrating cervical cancer screening with other healthcare services.

## Implications for research

The review suggests that integrating CCS can improve access to screening and increase uptake among women. Further research is needed to understand the factors contributing to the acceptability of integrated services and develop strategies to enhance acceptance. The review also emphasizes the benefits of integrating CCS services and, challenges such as inadequate funding, staffing issues, lack of equipment and medication, poor infrastructure, and fragmented services pose barriers to integration. Overall, the findings highlight the barriers and facilitators influencing the integration of CCS in low- and middle-income countries (LMICs) and can guide future research in developing strategies to overcome these barriers and promote integration in LMIC health institutions.

## Implications for policy makers

The study suggests that policy makers should utilize existing facilities and infrastructure for integrating cervical cancer screening services with reproductive and sexual health clinics, maternal health clinics, and well-baby clinics. They should also consider integrating screening with other healthcare services such as routine immunizations and family planning programs, as it can lead to cost savings. Additionally, policy makers need to address barriers related to the lack of policies on preventive oncology, insufficient funding, and limited coordination between healthcare sites. Overall, integrating cervical cancer screening is feasible and offers significant benefits, and policy makers should consider these findings to ensure successful integration and maximize women's health.

## Strength and limitations of the study

This is a novel scoping review that maps the literature on the integration of cervical cancer screening into health care facilities and its associated factors in LMICs. Besides, the study used five electronic databases containing peer-reviewed literature. This review is limited to publications in selected languages included in the review; this may bias some of the studies that maybe published in other language not included in this review. Inherent methodological limitation which the review included articles with poor quality assessment.

## Conclusion

In summary, the findings of the scoping review demonstrated that CCS integration into healthcare facilities in LMICs is possible, beneficial, and generally well-accepted by women. Integration within existing clinics offers opportunities for improved access, early detection, and cost efficiencies. This review also highlighted various barriers and facilitators across individual/women, provider, organizational/system, and policy levels for the uptake and integration of cervical cancer screening services. By addressing these challenges, healthcare systems, policymakers, and service providers can work collaboratively to increase participation in screening programs, improve screening services, and ultimately reduce the burden of cervical cancer. Efforts should focus on enhancing access, awareness, affordability, reducing stigma, providing comprehensive care, ensuring sufficient resources and staffing, and establishing supportive policies to ensure successful integration, optimal patient outcomes, and effectively address the barriers identified in this review. Further research and implementation of integrated CCS services are necessary to prioritize women's health in LMICs.

## Supporting information

**S1 Checklist. Preferred Reporting Items for Systematic Reviews and Meta-Analyses extension for Scoping Reviews (PRISMA-ScR) checklist.**
(DOCX)

**S1 Table. Search terms based on the population, concept, and context (PCC) framework for MEDLINE.**
(DOCX)

**S2 Table. Joanna Briggs Institute (JBI) quality appraisal tool.**
(DOCX)

## Author Contributions

**Conceptualization:** Rahel Nega Kassa.

**Data curation:** Rahel Nega Kassa, Desalegn Markos Shifti, Kassahun Alemu, Akinyinka O. Omigbodun.

**Formal analysis:** Rahel Nega Kassa.

**Funding acquisition:** Rahel Nega Kassa.

**Investigation:** Rahel Nega Kassa.

**Methodology:** Rahel Nega Kassa, Desalegn Markos Shifti, Akinyinka O. Omigbodun.

**Project administration:** Rahel Nega Kassa.

**Resources:** Rahel Nega Kassa.

**Software:** Rahel Nega Kassa, Desalegn Markos Shifti.

**Supervision:** Kassahun Alemu, Akinyinka O. Omigbodun.

**Validation:** Rahel Nega Kassa, Desalegn Markos Shifti, Kassahun Alemu, Akinyinka O. Omigbodun.

**Visualization:** Rahel Nega Kassa, Desalegn Markos Shifti, Kassahun Alemu, Akinyinka O. Omigbodun.

**Writing – original draft:** Rahel Nega Kassa.

**Writing – review & editing:** Rahel Nega Kassa, Desalegn Markos Shifti, Kassahun Alemu, Akinyinka O. Omigbodun.

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
