## [Decision Letter · Decision Letter 0]

7 Feb 2024

PGPH-D-23-02587

Integration of cervical cancer screening into healthcare facilities in low-and middle-income countries: a scoping review

Dear Dr. Kassa,

Thank you for submitting your manuscript to PLOS Global Public Health. After careful consideration, we feel that it has merit but does not fully meet PLOS Global Public Health’s publication criteria as it currently stands. Therefore, we invite you to submit a revised version of the manuscript that addresses the points raised during the review process.

We look forward to receiving your revised manuscript.

Kind regards,

Sadeep Shrestha

Academic Editor

Journal Requirements:

1. We noticed you have some minor occurrence of overlapping text with the following previous publication(s), which needs to be addressed:

- https://doi.org/10.1002/ijgo.12194

-https://doi.org/10.1186/s12913-023-09326-6

In your revision ensure you cite all your sources (including your own works), and quote or rephrase any duplicated text outside the methods section. Further consideration is dependent on these concerns being addressed.

If you did not receive any funding for this study, please simply state: “The authors received no specific funding for this work.

2. Please provide separate figure files in .tif or .eps format only and remove any figures embedded in your manuscript file. Please also ensure all files are under our size limit of 10MB.

Additional Editor Comments (if provided):

Reviewers' comments:

Reviewer's Responses to Questions

**Comments to the Author**

1. Does this manuscript meet PLOS Global Public Health’s publication criteria? Is the manuscript technically sound, and do the data support the conclusions? The manuscript must describe methodologically and ethically rigorous research with conclusions that are appropriately drawn based on the data presented.

Reviewer #1: Partly

Reviewer #2: Yes

2. Has the statistical analysis been performed appropriately and rigorously?

Reviewer #1: Yes

Reviewer #2: Yes

3. Have the authors made all data underlying the findings in their manuscript fully available (please refer to the Data Availability Statement at the start of the manuscript PDF file)?

Reviewer #1: Yes

Reviewer #2: Yes

4. Is the manuscript presented in an intelligible fashion and written in standard English?

Reviewer #1: Yes

Reviewer #2: Yes

5. Review Comments to the Author

Reviewer #1: This is nicely written and thorough review of the integration of cervical cancer screening into existing healthcare facilities in LMICs. As you will see in my review there are 2 major concerns: lack of clarity into the type of healthcare facility you are interested in (or is it all off them) and the manuscript is too long and repetitive.

You are interchanging and mixing “primary care facilities” and “healthcare facilities” which is causing confusion about what facilities you mean and are interested in. I think what you mean is: investigating CC screening into existing healthcare facilities that offer primary care services rather than offering CC screening as stand-alone services. This needs to be clarified early on (Introduction, line 63 would be a good place). Then be sure to choose and use the same language throughout (such as “healthcare facilities”). The abstract seem to imply you were comparing primary care facilities vs. all others. But Methods imply the type of facility does not matter.

Abstract:

Last sentence that “addressing the gaps could promote…integration..into primary care services”. Are you considering the various settings listed in the sentence prior and the results to be “primary care services”? If yes, please clarify earlier. If not, then rephrase this sentence because the earlier presented information indicates it is successful in many settings.

Introduction:

Last paragraph: you state there are several studies on CC screening uptake but there is still room for more comprehensive evidence. Please elaborate on WHY there is still a need for comprehensive evidence? As stated, there is no gap for you to address…

Methods:

Line 53-54: Is this references Figure 1? If so, mention that.

Results:

Overall the Results section while nicely written is too long and repetitive. It reads like a Discussion section. To cut: (1) you should only present facts directly from the studies reviewed and remove opinions/conclusions/narrative from yourself—this is done in the Discussion. For example: Line 227: after “highlighting” is your conclusion; (2) Cut language. Simply state the fact as it relates to the Subheading without additional descriptive language (Example: Line 329-330 you do not need. Jump right into the barriers. A suggestion to consider is using bullet points in each subheading to briefly state each relevant finding; and (3) consider combining subheads where information is repetitive or could go together.

I suggest cutting the Sections on Lines 212-225. The first is not needed and the second is already covered by your table which shows the types of facilities and you describe the facilities you found in the first 2 paragraphs. Everything here is repetitive.

Line 252-253: Please add the reference for this study.

Many of the results in each section are related to HIV clinics or HIV care, etc. There were many other clinics with CCS integration. Please make sure to mention those results. When you discuss HIV only clinics make sure the text is all together in that section. For example in Section: Benefits of Integration of CCS Service into Healthcare Facilities...Please consider re-arranging this section to flow better. You present several points specific to HIV clinics and then mix in statements related to other types of clinical outside of HIV. I suggest trying a paragraph on the benefits to integrating into HIV clinics and then another on the non-HIV specific information.

Numbers and statistics should be provided to support statements throughout. For example, Line 238 you say “most women included accepted…”. Please quantify “most” (i.e. 87% of women)

Discussion

This section is nicely written and what I expect from a Discussion. But with the current Results section it simply repeats. Making changes to the Results will help.

Reviewer #2: Summary

The study addresses a clearly important public health problem and explorers how the World Health Organization recommendations on integration of cervical cancer screening in health facilities are being implemented as well as facilitators and barriers. The goal of screening for cervical cancer is to find precancerous cervical cell changes when treatment can prevent cervical cancer from developing. The article is technically sound, has methodological rigor and is presented in an intelligible fashion.

Strengths

1. Use of multiple databases: Five electronic databases with peer-reviewed literature were used.

2. Scooping review conducted quality assessment of the journal articles.

3. the data supports most of the conclusions and is available in the manuscript

Weaknesses

1. While the author identifies the burden by region, the author fails to link the high burden of Cervical cancer in SSA to the persistent infection with the human papillomavirus (HPV), regional differences in the cervical cancer burden are related to inequalities in access to vaccination, screening and treatment services, risk factors including HIV prevalence, and social and economic determinants such as sex, gender biases and poverty and the fact that women living with HIV are 6 times more likely to develop cervical cancer compared to women without HIV.

2. While the scooping review conducted quality assessment of the journal articles, those very poorly on quality were also included (inherent weakness of scooping reviews)

3. In the discussion section, the authors could increase relevance of the discussion by including contemporary guidance such as and the recent WHO recommendations on HPV DNA testing. Linking study finding and the literature with the contemporary recommendations for screening could increase the utility of the research findings and implications for scale up

4. In line 464, there is a mention for further research needed to understand the factors contributing to the acceptability of integrated services and develop strategies to enhance acceptance. Most (all) of the studies in the scooping review considered VIA as the main screening method for cervical cancer. With the recommendation on integration of HPV testing as part of the screening algorithms, the study does not provide insights on how this could impact the conclusions. This could be another area of further research

Recommendation:

I recommend approval with minor changes.

Evidence and Examples

Major issues:

1. Beyond the targets for Cervical cancer control by 2030 and the 90-70-90 targets; it is advisable that the authors also provide recent literature of the screening recommendations by WHO on Cervical cancer screening using DNA PCR. In one or another, inclusion of the recent WHO recommendations of HPV DNA PCR testing could have an implication on the policy considerations in the conclusion of this study (lines 491-503)

2. While quality of the articles was assessed, the analysis includes results from some of the poor-quality articles. However, this inherent methodological limitation/observation is not included in the limitation of the study.

3. In the discussions, provide insight into regional variations of cervical cancer burden and screening practices and the findings in the scooping review.

Minor issues:

4. Line 99 “published from were retrieved the year 2000’ has a typographical error caused by sequencing of words. Please correct to “ published were retrieved from the year….

5. Lines 218 and 227 refer to most women. Kindly quantify the word most by providing the ratio or percentage in the brackets to validate the choice of the word most

6. Please clarify: Line 301/302 does not show integration, and yet presented as a barrier to uptake.

7. Please clarify: Line 330/331 refers to fragmented services while referring to integration. Kindly clarify

Other Points

-None

6. PLOS authors have the option to publish the peer review history of their article (what does this mean?). If published, this will include your full peer review and any attached files.

**Do you want your identity to be public for this peer review?** For information about this choice, including consent withdrawal, please see our Privacy Policy.

Reviewer #1: No

Reviewer #2: No

While revising your submission, please upload your figure files to the Preflight Analysis and Conversion Engine (PACE) digital diagnostic tool, https://pacev2.apexcovantage.com/. PACE helps ensure that figures meet PLOS requirements. To use PACE, you must first register as a user. Registration is free. Then, login and navigate to the UPLOAD tab, where you will find detailed instructions on how to use the tool. If you encounter any issues or have any questions when using PACE,

---

## [Decision Letter · Decision Letter 1]

12 Apr 2024

Integration of cervical cancer screening into healthcare facilities in low-and middle-income countries: a scoping review

PGPH-D-23-02587R1

Dear Assistant professor Kassa,

We are pleased to inform you that your manuscript 'Integration of cervical cancer screening into healthcare facilities in low-and middle-income countries: a scoping review' has been provisionally accepted for publication in PLOS Global Public Health.

Best regards,

Sadeep Shrestha

Academic Editor

Reviewer Comments (if any, and for reference):

Reviewer's Responses to Questions

**Comments to the Author**

1. If the authors have adequately addressed your comments raised in a previous round of review and you feel that this manuscript is now acceptable for publication, you may indicate that here to bypass the “Comments to the Author” section, enter your conflict of interest statement in the “Confidential to Editor” section, and submit your "Accept" recommendation.

Reviewer #1: All comments have been addressed

Reviewer #2: All comments have been addressed

2. Does this manuscript meet PLOS Global Public Health’s publication criteria? Is the manuscript technically sound, and do the data support the conclusions? The manuscript must describe methodologically and ethically rigorous research with conclusions that are appropriately drawn based on the data presented.

Reviewer #1: Yes

Reviewer #2: Yes

3. Has the statistical analysis been performed appropriately and rigorously?

Reviewer #1: N/A

Reviewer #2: Yes

4. Have the authors made all data underlying the findings in their manuscript fully available (please refer to the Data Availability Statement at the start of the manuscript PDF file)?

Reviewer #1: Yes

Reviewer #2: Yes

5. Is the manuscript presented in an intelligible fashion and written in standard English?

Reviewer #1: Yes

Reviewer #2: Yes

6. Review Comments to the Author

Reviewer #1: (No Response)

Reviewer #2: (No Response)

7. PLOS authors have the option to publish the peer review history of their article (what does this mean?). If published, this will include your full peer review and any attached files.

**Do you want your identity to be public for this peer review?** For information about this choice, including consent withdrawal, please see our Privacy Policy.

Reviewer #1: No

Reviewer #2: **Yes: **Dr Samson Haumba
